**Data Availability Statement:** All sequence data generated in this study are deposited in the Sequence Read Archive under the BioProject

# Malian children infected with *Plasmodium ovale* and *Plasmodium falciparum* display very similar gene expression profiles

Kieran Tebben[1,2], Salif Yirampo[3], Drissa Coulibaly[3], Abdoulaye K. Koné[3], Matthew B. Laurens[4], Emily M. Stucke[4], Ahmadou Dembélé[3], Youssouf Tolo[3], Karim Traoré[3], Amadou Niangaly[3], Andrea A. Berry[4], Bourema Kouriba[3], Christopher V. Plowe[4], Ogobara K. Doumbo[3†], Kirsten E. Lyke[4], Shannon Takala-Harrison[4], Mahamadou A. Thera[3], Mark A. Travassos[4], David Serre[1,2]*

1 Institute for Genome Sciences, University of Maryland School of Medicine, Baltimore, Maryland, United States of America, 2 Department of Microbiology and Immunology, University of Maryland School of Medicine, Baltimore Maryland, United States of America, 3 Malaria Research and Training Center, University of Sciences, Techniques and Technologies, Bamako, Mali, 4 Malaria Research Program, Center for Vaccine Development and Global Health, University of Maryland School of Medicine, Baltimore, Maryland, United States of America

† Deceased.
* dserre@som.umaryland.edu

## Abstract

*Plasmodium* parasites caused 241 million cases of malaria and over 600,000 deaths in 2020. Both *P. falciparum* and *P. ovale* are endemic to Mali and cause clinical malaria, with *P. falciparum* infections typically being more severe. Here, we sequenced RNA from nine pediatric blood samples collected during infections with either *P. falciparum* or *P. ovale*, and characterized the host and parasite gene expression profiles. We found that human gene expression varies more between individuals than according to the parasite species causing the infection, while parasite gene expression profiles cluster by species. Additionally, we characterized DNA polymorphisms of the parasites directly from the RNA-seq reads and found comparable levels of genetic diversity in both species, despite dramatic differences in prevalence. Our results provide unique insights into host-pathogen interactions during malaria infections and their variations according to the infecting *Plasmodium* species, which will be critical to develop better elimination strategies against all human *Plasmodium* parasites.

## Author summary

Multiple species of *Plasmodium* parasites can cause human malaria. Most studies and elimination efforts target *P. falciparum*, the most common cause of malaria worldwide and the species responsible for the vast majority of the mortality. Other *Plasmodium* species, such as *P. ovale*, typically lead to less severe forms of the disease but little is known about the molecular mechanisms at play during malaria infections with different parasites. We analyzed host and parasite gene expression from children successively infected with

PRJNA878485. Custom scripts are available at https://github.com/tebbenk/PfPo_RNAseq.

**Funding:** This work was supported by awards from the National Institutes of Health (R21AI146853 to DS and MAT and R01HL146377 to MAT) and an NIAID-funded predoctoral fellowship (T32 AI095190 to KT). Participant enrollment and sample collection were supported by NIH grants U01AI065683, R01HL130750 and D43TW001589 to CVP and R01AI099628 to MATh. The funders had no role in study design, data collection and analysis, decision to publish, or preparation of the manuscript.

**Competing interests:** None

*P. ovale* and *P. falciparum* and found that, while the parasite gene expression differed significantly, the transcriptional profiles of the host immune cells were similar in *P. ovale* or *P. falciparum* infections. This suggests that infected individuals respond to uncomplicated malaria infections similarly, regardless of the *Plasmodium* species causing the infection, and that alternative immune processes may become important during the progression to severe *P. falciparum* malaria rather than being inherent features of *P. falciparum* infections. Additionally, we observed similar levels of genetic diversity among *P. ovale* and *P. falciparum* parasites, suggesting that the *P. ovale* population might be larger than currently thought, possibly due to extensive misdiagnosis or the existence of hidden reservoirs of parasites.

## Introduction

*Plasmodium* parasites caused 241 million cases of malaria and over 600,000 deaths in 2020 [1], a partial reversal of decades of progress towards elimination. Malaria symptoms derive from the asexual replication of *Plasmodium* parasites in human red blood cells (RBCs)[2]. At least five species of *Plasmodium* parasites commonly cause human malaria: *P. falciparum*, *P. vivax*, *P. ovale*, *P. malariae* and *P. knowlesi* [2]. *P. falciparum* is responsible for the majority of infections worldwide and is the dominant species in Sub-Saharan Africa [2,3]. *P. falciparum* can cause severe malaria [2] and is responsible for the majority of malaria deaths due to complications such as severe anemia or cerebral malaria. This high pathogenicity is thought to be, at least partially, due to the sequestration of mature asexual *P. falciparum* parasites in the microvasculature [4]. Post-mortem analyses of brain [5] and kidney [6] from individuals with severe malaria show parasite accumulation in the tissue microvasculature. This accumulation can lead to obstructions and focal hypoxia, and local increases in inflammatory cells and molecules responding to the parasites [5]. Rupture of adhered infected RBCs after schizont maturation can also lead to focal release of parasite antigens and immune-activating factors, contributing to localized tissue damage at the site of adherence [6]. While most studies focus on *P. falciparum*, other *Plasmodium* species also cause significant public health burden and the lack of specific knowledge about these parasites is becoming increasingly problematic. *P. vivax* is common in South Asia and South America but rare in Sub-Saharan Africa [1,7], while *P. knowlesi* is a recent zoonotic parasite causing infections in Southeast Asia [8]. *P. ovale* and *P. malariae* are widely distributed parasites that have typically been considered as relatively rare [9] and causing milder infections than *P. falciparum* [3,10,11]. Infection with these parasites typically leads to lower fevers [10] and parasitemia [10,11] than those with *P. falciparum*, possibly due to a slower intraerythrocytic replication with fewer merozoites produced per cycle [11] and a preference for specific RBCs [10,11]. Because these parasites are difficult to detect on peripheral blood smear [10,11] and often occur in coinfections with more virulent species [9], they are likely underdiagnosed [9]. *P. ovale* is further categorized into two phenotypically indistinguishable sub-species or species [12], *P. ovale curtisi* and *P. ovale wallikeri*, that are often co-endemic, notably in West Africa [2,3,13].

Since blood-stage parasites play a central role in disease severity, transmission and immunity/immune-evasion, analyses of host and parasite gene expression from infected blood samples could provide critical information on these processes and their regulation. Such studies have been performed from *P. falciparum* infections, either separately to examine changes in host [14–16] or parasite [17–22] transcriptional regulation during infections, or jointly [23–26] from the same samples to investigate host-pathogen interactions. By contrast, a single study has

characterized the gene expression profiles of *P. ovale* parasites during an infection [27], and none have examined the host transcriptome during infections with this parasite.

Here, we used dual RNA-sequencing (dual RNA-seq) to examine whole blood samples from three Malian children successively infected with *P. falciparum* and *P. ovale*, and simultaneously characterized the gene expression profiles of the host and parasites. Our analyses provide novel transcriptomic and genetic insights on *P. ovale* infections and allow a first examination of the host immune response to these infections and how this response differs from the immune response to *P. falciparum* infections.

## Results

### Characterization of host and parasite gene expression profiles by dual RNA-seq

We extracted and sequenced RNA from nine blood samples collected from three Malian children successively infected with *P. falciparum* and *P. ovale* (**Table 1**). All samples were collected during a patient-initiated, unscheduled visit in response to self-assessed malaria symptoms (i.e., fever, headaches, joint pain, vomiting, diarrhea, or abdominal pain) and with the presence of malaria parasites confirmed by microscopy [28]. To confirm the *Plasmodium* species detected by microscopy [28], we simultaneously mapped all reads to the genomes of *P. falciparum*, *P. malariae*, *P. ovale* and *P. vivax*. In each sample, more than 98% of the *Plasmodium* reads mapped to the species identified by thick smear microscopy, with the exception of one sample (C2) for which 93.9% of the *Plasmodium* reads mapped to *P. ovale* (the species determined by microscopy) and 5.4% of the reads mapped to *P. falciparum*, possibly suggesting a co-infection (**S1 Table**).

To analyze the gene expression profiles of the host and parasites in each infection, we then mapped all reads simultaneously to the *P. falciparum*, *P. ovale* and human genomes (**S2 Table**). Overall, for each sample, we obtained more than 48 million reads (70–92%) mapped to the human genome and more than 1 million reads (2–25%) mapped to the *Plasmodium* genome, providing a robust characterization of the host and parasite gene expression profiles of each infection. (Note that due to the high proportion of PCR duplicates (>94%), we excluded sample B4 from all analyses presented below (**S2 Table**).)

**Table 1. Epidemiological and clinical characteristics of study participants.**

| ID | Collection Month | Collection season | Sex | Ethnicity | Age (years) | Species | Parasitemia (parasites/µL blood) | Temperature (°C) | Hgb con. (g/dL) |
|----|------------------|-------------------|-----|-----------|-------------|---------|----------------------------------|------------------|-----------------|
| A1 | October | Wet | F | Dogon | 5 | *P. falciparum* | 16,950 | 36 | 8.9 |
| A2 | February | Dry | | | 7 | *P. ovale* | 2,025 | 38.1 | 8.9 |
| B1 | March | Dry | F | Bambara | 9 | *P. ovale* | 11,375 | 37.8 | 12.4 |
| B2 | May | Dry | | | 9 | *P. ovale* | 800 | 39.5 | 11.5 |
| B3 | September | Wet | | | 8 | *P. falciparum* | 600 | 37.2 | 11.9 |
| B4* | November | Wet | | | 8 | *P. falciparum* | 18,300 | 37.4 | 12 |
| C1 | December | Wet | M | Dogon | 12 | *P. falciparum* | 131,700 | 38 | 10.9 |
| C2 | March | Dry | | | 13 | *P. ovale* | 4,825 | 38.9 | 10.9 |
| C3 | September | Wet | | | 13 | *P. falciparum* | 11,950 | 38.2 | 11 |

* Sample B4 was excluded from all analyses because of a high proportion of PCR duplicate reads.

## Host gene expression varies more between individuals than between infecting species

We first used principal component analysis (PCA) to examine the relationships among the host gene expression profiles characterized from each *P. falciparum* and *P. ovale* infection. Interestingly, the host gene expression profiles seemed to cluster according to the participant rather than by the infecting parasite species (**Fig 1A**). To rigorously quantify this observation, we calculated the proportion of the variance in gene expression [29] explained by the infecting parasite species, the parasitemia, the age of the participant at the time of the infection, and inter-individual differences (**Fig 1B**). Consistent with the PCA, the infecting parasite species explained less than 5% of the variance in host gene expression (median: 0%, IQR: 0%– 2.2%). By contrast, the child's age at the time of infection explained, on average, a third of the variance (median: 34%, IQR: 11.1%– 57%). Note that given the small number of samples analyzed here, it was difficult to rigorously determine the variance in gene expression explained by inter-individual differences vs. age (or sex and ethnicity that cannot be considered here) as these variables are confounded in our sample. However, these analyses clearly showed that the variance in host gene expression during a *Plasmodium* infection was primarily driven by host factors and that the infecting parasite species had, comparatively, little effect.

## The few host genes differentially expressed between *P. falciparum* and *P. ovale* infections are involved in regulation of adaptive immunity

We then tested whether specific host genes were differentially expressed between *P. ovale* and *P. falciparum* infections. To account for the important effects of inter-individual differences on host gene expression, we used a paired design for these statistical analyses: we selected two sequential samples per individual, one infected with *P. falciparum* and one infected with *P. ovale*, to minimize differences in the age of the participant and other inter-individual factors (**S1 Fig**). Consistent with the apparent similarity of host expression during *P. ovale* and *P.*

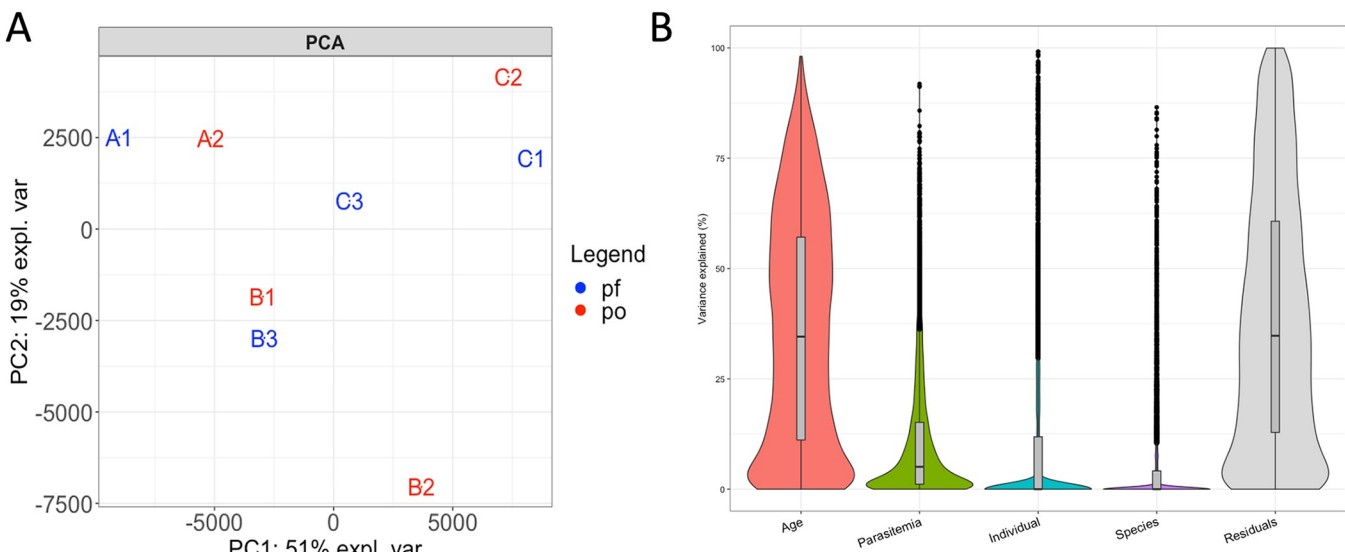

**Fig 1. Host gene expression differs more between individuals than according to the infecting species.** (**A**) PCA showing the relationships among the eight infections based on the expression level of 9,884 human genes and colored according to the infecting species (blue–*P. falciparum*, red–*P. ovale*). (**B**) Percentage of the variance in host gene expression explained by the age of the child, the parasitemia of the infection, the individual and the infecting species.

*falciparum* infections (**Fig 1**), only 127 host genes out of 9,884 genes tested, were deemed significantly differentially expressed between *P. ovale* and *P. falciparum* infections (FDR < 0.1, **S1A Fig** and **S3 Table**), compared to more than 1,500 differentially expressed parasite genes (see below). To understand whether differences in host gene expression were a result of i) true differences in gene regulation or ii) possible differences in immune cell composition, we used CIBERSORTx [30] to estimate, directly from the RNA-seq reads, the relative proportion of each immune cell subset present in each *P. falciparum*- and *P. ovale*-infected sample (**S1B Fig**). As expected given the similarity in host gene expression profiles, we did not detect any significant differences in immune cell composition between the *P. falciparum* and *P. ovale* infections in any of the three individuals (chi-squared test, p > 0.1), although it is important to note that this analysis may not be able to detect minor, but potentially important, differences in immune cell composition. This overall similarity in relative proportions of immune cell populations during *P. ovale* and *P. falciparum* infections might be surprising given the differences in disease severity caused by each parasite species [3]. Since all blood samples analyzed here were collected from uncomplicated malaria infections with relatively similar presentation (e.g., temperature, hemoglobin level and clinical assessment, **Table 1**) [28], these findings could suggest that alternative immune processes and immune cell recruitment may become important during the progression to severe *P. falciparum* malaria, rather than being inherent features of all *P. falciparum* infections.

Among the host genes with significantly higher expression in *P. ovale* infections than in *P. falciparum* infections, we observed the presence of genes involved in the activation of the innate immune system, such as complement proteins (C1QA, C1QB and C1QC) [31] and genes involved in the activation of the NLRP3 inflammasome (GBP5[32]) or antigen presentation (HLA-DRB1 [33], HLA-DMB [33], HLA-DMA [33], HLA-DRA [33], CD40 [34]) (**S1A Fig** and **S3 Table**). Additionally, we observed, in *P. ovale* infections, increased expression of genes involved in the suppression of T-cell mediated responses to pathogens, such as IL-18BP [35], IDO1 [36,37], IL27 [38–40], and SOCS1 [41,42] (**S1A Fig** and **S3 Table**). By contrast, several genes related to dendritic cell development, a particularly important cell type for bridging the innate and adaptive immune responses, showed significantly higher expression in *P. falciparum* compared to *P. ovale* infections (**S1A Fig** and **S3 Table**). For example, WLS has been reported to be essential for dendritic cell homeostasis [43] and TSPAN13 is highly expressed in plasmacytoid dendritic cells [44].

## *Plasmodium* gene expression differs more by species than by infected host

RNA-seq data generated directly from infected blood samples enables simultaneous characterization of host and parasite gene expression and we next focused on examining differences in parasite gene expression. To compare the gene expression of *P. falciparum* and *P. ovale* parasites during symptomatic infections, we only included 2,631 expressed genes that had one-to-one orthologs in both species (see Material and Methods). Again, we first used PCA to investigate how global gene expression differs between uncomplicated infections with *P. falciparum* and *P. ovale*. In contrast to host gene expression, the parasite gene expression profiles from each species were clearly separated by PC1 (**Fig 2A**), with *P. ovale* infections clustering closely together and the *P. falciparum* samples spread along PC2. Indeed, on average one third of the variance in parasite gene expression was explained by the parasite species (median: 39.1%, IQR: 10.8% - 61.6%), while the age of the child explained 14.6% of the variance, and the parasitemia and individual differences less than 6%, on average (**Fig 2B**). While not surprising, these findings are in stark contrast with the patterns observed for the host gene expression profiles

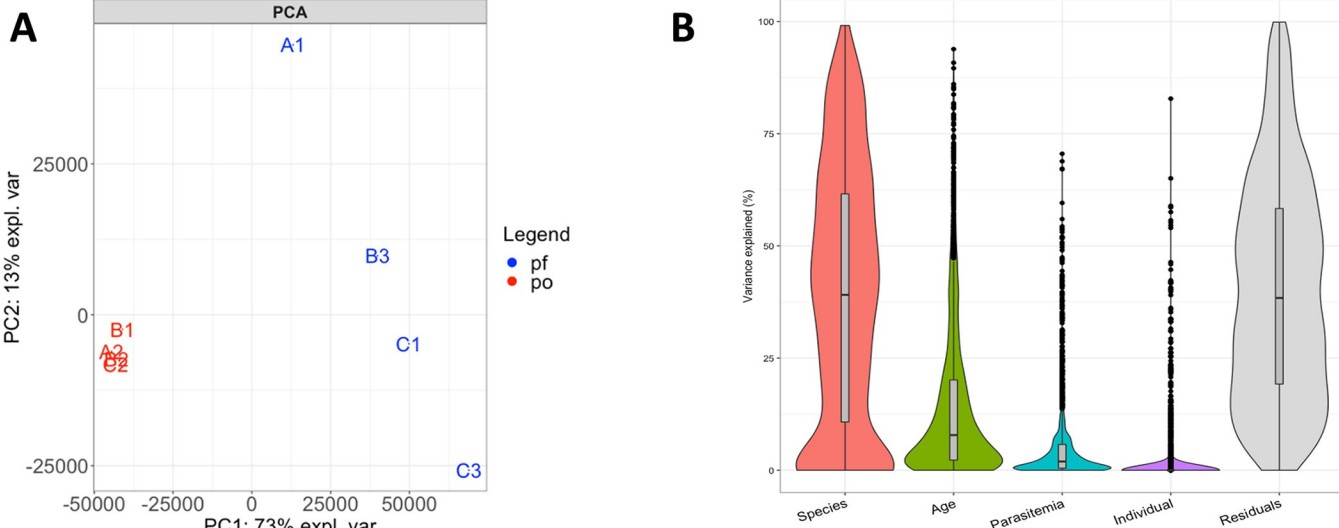

**Fig 2. *Plasmodium* gene expression profiles during symptomatic infections cluster according to the infecting species. (A)** PCA showing the relationships among the eight infections based on the expression level of 2,631 *Plasmodium* genes and colored according to the infecting species (blue–*P. falciparum*, red–*P. ovale*). (**B**) Percentage of the variance in parasite gene expression explained by the infecting species, the age of the child, the parasitemia of the infection and the individual.

of the same infections, suggesting that, while inducing a relatively similar response of the host, *P. falciparum* and *P. ovale* parasites are regulated differently in the blood.

### Differences in parasite gene expression in *P. ovale* and. *P. falciparum* infections are largely explained by differences in stage composition

We then statistically tested which orthologous parasite genes were differentially expressed in *P. falciparum* vs. *P. ovale* infections, and since host factors (i.e., immunity, age, sex) seemed to contribute little to the variations in parasite gene expression, we included all eight samples from both species in our statistical analyses. We identified 1,858 parasite genes that were differentially expressed according to the parasite species (while only 127 host genes were differentially expressed) (FDR < 0.1, **S2A Fig** and **S4 Table**). Note that reducing the sample size to the same six infections used in the host expression analysis, and using a paired analysis framework, only minimally affected these results, with 1,624 differentially expressed parasites genes (**S2D Fig**).

To understand whether these differences were the result of i) true differences in gene expression or ii) differences in the relative proportion of the various parasite developmental stages present in each blood sample, we used a species-agnostic gene expression deconvolution [45] to estimate the stage composition of each *P. falciparum* and *P. ovale* sample (**S2C Fig**). The stage compositions of *P. falciparum* infections appeared more variable than those of *P. ovale* infections, with *P. ovale* infections consistently displaying a majority of ring-stage parasites across all samples (mean = 57%), while *P. falciparum* infections showed variable proportions of trophozoites (**S2C Fig**). In fact, the relative proportion of trophozoites in *P. falciparum* infections was significantly correlated with the overall variation in gene expression captured by PC2 (p = 0.043, **S3 Fig**), suggesting that variations in stage composition among samples drive some of the differential gene expression observed. The stage composition of these infections is perhaps surprising since one would expect *P. falciparum* ring-stage parasites to be highly predominant in peripheral blood (since mature *P. falciparum* blood stages usually

sequester in tissues [4]), while sequestration has not been described for *P. ovale* [10]. Note however that the stage composition estimated by gene expression deconvolution reflects the proportion of transcripts derived from each stage and, since rings are less transcriptionally active than other blood stages [20], the proportion of ring stage parasites is systematically (but proportionally) underestimated. Additionally, it is possible that the inference of the stage composition did not work as well for *P. ovale* since its stage-specific gene expression profile remain incompletely characterized (although this gene expression deconvolution strategy has been validated on divergent *Plasmodium* species [45]).

After accounting for stage composition differences, we observed a dramatic reduction in the number of differentially expressed genes between parasite species (from 1,858 to 118, **S2B Fig** and **S4 Table**), confirming that stage composition differences explained the vast majority of parasite gene expression differences observed between *P. ovale* and *P. falciparum* infections. A few genes associated with gametocyte function remained significantly more expressed in *P. ovale* vs. *P. falciparum* infections (e.g., G377 [46], CRISP [47], and PM6 [48]*)*, possibly suggesting differential regulation of the parasite sexual stages between species. Conversely, a putative homolog of T-cell immunomodulatory protein (TIP), which has been described to have anti-inflammatory effects in *P. berghei* models [49], was expressed at higher levels in *P. falciparum* infections. However, given the small sample size of this study, the analyses of individual gene expression should be interpreted with caution and further studies will be required to confirm these findings.

## RNA-seq data provide a preliminary assessment of the genetic diversity of *P. falciparum* and *P. ovale* symptomatic infections in Mali

Since RNA-seq provides a characterization of the gene expression by sequencing mRNA molecules, one can leverage these data to examine genetic variants located in the expressed transcripts. Between 3,027,680 and 10,274,779 nucleotide positions of the *Plasmodium* genome were sequenced at >20X in each sample, allowing a robust investigation of the parasite genetic diversity (**S2 Table**).

We first examined whether we could detect allelic variations within each sample, which would indicate the presence of multiple genetically different parasites in the circulation. At all nucleotide positions sequenced, we only observed one allele for all *P. ovale* isolates, while the data suggested that some of the *P. falciparum* infections may contain one additional (but rare) clone (**S4 Fig**).

We next focused on genetic differences between parasites from different infections. Since two sub-species of *P. ovale* can cause human malaria [12], we reconstructed, directly from the RNA-seq data, the entire cytochrome B gene sequence from each *P. ovale* infection and compared them with published sequences for both *P. ovale curtisi* and *P. ovale wallikeri*. The sequences generated from the infections A2, B1 and B2 clearly clustered with *P. ovale curtisi* sequences, while the sequence reconstructed from infection C2 clustered with *P. ovale wallikeri* sequences (**Fig 3A**). Interestingly, despite the ancient divergence of the two *P. ovale* sub-species [9,12], the parasite (**Fig 2**) or host (**Fig 1**) gene expression profiles generated from these infections were undistinguishable when analyzed with *P. falciparum* infections, suggesting that both subspecies were similarly regulated in blood infections and had, overall, similar consequences on the host gene expression. (Note that when only *P. ovale* infections were considered, the gene expression profiles derived from the *P. ovale wallikeri* infection seemed to be distinct from those generated from *P. ovale curtisi* (**S5 Fig**), although the small number of samples analyzed prevented any definitive conclusion.

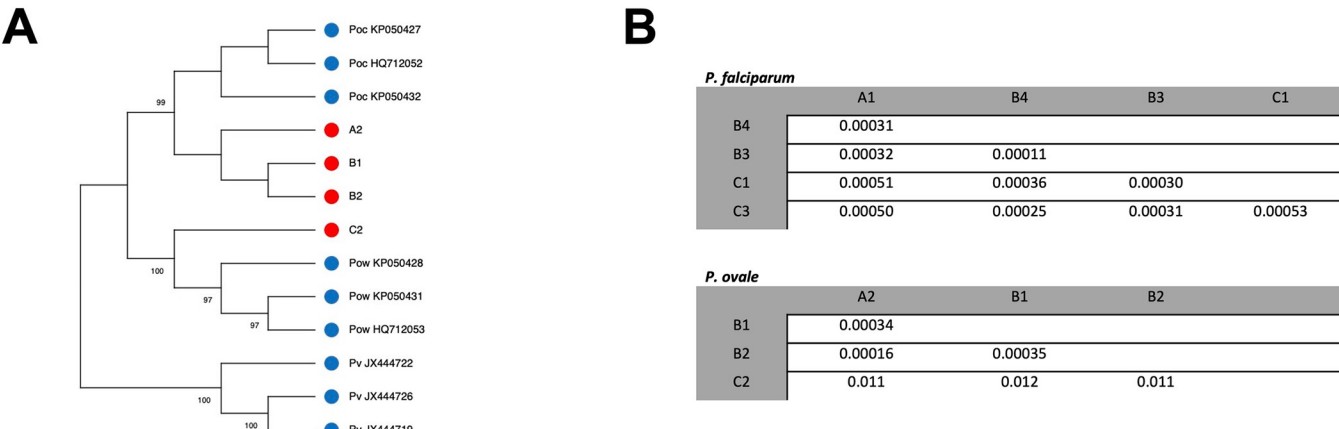

**Fig 3. Genetic relationships among the *Plasmodium* parasites analyzed by RNA-seq. (A)** Neighbor-joining tree showing the relationships among cytochrome B sequences from the *P. ovale* parasites studied in this study (red) and sequences from NCBI (blue). Note that the sequences from the A2, B1 and B2 infections cluster with *P. ovale curtisi* sequences while the sequence from infection C2 clusters with *P. ovale wallikeri* sequences. The number next to each branch indicates the percentage of replicate trees in which the sequences clustered together based on 500 bootstraps. **(B)** Proportion of pairwise nucleotide differences between pairs of *P. falciparum* (top) and *P. ovale* (bottom) infections based on positions covered at >20X. Note the higher proportion of pairwise nucleotide differences in pairs including the C2 infection, consistent with its *P. ovale wallikeri* determination.

We then examined genome-scale differences in genetic diversity among the *P. falciparum* and *P. ovale* isolates by calculating the proportion of pairwise nucleotide differences between each pair of samples of the same species. While both *Plasmodium* species are endemic in Mali [50], *P. falciparum* is much more prevalent than *P. ovale* [51] and one might therefore have expected a higher diversity among *P. falciparum* parasites. However, on average the proportion of differences was very similar between *P. falciparum* (mean: 0.00035, sd: 0.00013) and *P. ovale curtisi* (mean: 0.00028, sd: 0.00011) (t-test p-value = 0.44) isolates (**Fig 3B**) suggesting that despite their likely smaller population size, *P. ovale* parasites maintain a high genetic diversity.

## Discussion

Our data indicate that host gene expression does not differ dramatically between uncomplicated malaria infections caused by *P. falciparum* or *P. ovale*. This finding is somewhat surprising since species-specific immune responses during *Plasmodium* infections have been described in rodent [52,53] and human [54–56] studies. One possible explanation for this discrepancy is that previous studies compared infections with different severities and/or different infected individuals. The large differences in gene expression observed between infections caused by distinct *Plasmodium* species in those studies might therefore have been confounded by host gene expression differences associated with different disease presentations (e.g., severity or symptoms) and/or inter-individual variations. By contrast, our study compared the host gene expression profiles during similar uncomplicated malaria infections and in the same individuals, and clearly showed that host factors contribute more, quantitatively, to the host gene expression profiles during malaria infections than the infecting *Plasmodium* species (based on the number of differentially expressed human and *Plasmodium* genes). This result is consistent with reports of important interindividual differences [57] in the susceptibility to [58–60], and immune response against [61,62], *P. falciparum* infections. Alternatively, previous studies of host gene expression may have been confounded by differences in parasitemia since the parasite load typically differ between parasite species [3] and has been shown, within one species, to be associated with variations in host gene expression [26]. (Our analyses may also suffer

from differences in parasitemia between infections, but those variations are not entirely confounded with the infecting species in our sample).

We observed that the age of the individual significantly contributed to the differences in host gene expression. This observation could reflect the maturation of the immune system in young children [63], although, given our small sample size of this study, it is difficult to rigorously evaluate the individual contribution of different host factors (e.g., age, sex or ethnicity) which are confounded in this study. Future studies including more samples are needed to fully disentangle the role of these host factors, and other clinical variables, on the host gene expression during malaria infections. Despite the overwhelming importance of individual factors on the host gene expression, we detected statistically significant differences associated with the infecting species for a small number of human genes, possibly reflecting differences in the host response to these two *Plasmodium* species. We found a higher expression of genes related to dendritic cell development during *P. falciparum* infections, possibly influencing the effective bridging of the innate and adaptive immune system during infections by this species. *P. falciparum* parasites have been shown to lead to atypical activation of dendritic cells [64], but the comparison of dendritic cell responses to infection by different *Plasmodium* species may reveal important species-specific interactions. By contrast, we found that genes involved in activation of the innate immune system and T-cell suppression were expressed at higher levels in *P. ovale* infections, compared to *P. falciparum* infections. This is consistent with reports that, per parasitized RBC, *P. ovale* induces a stronger immune response than *P. falciparum* [65]. However, given our small sample size, statistical results for specific individual genes should be interpreted with caution.

In contrast to the human gene expression results, we found that parasite gene expression vastly differs by species. This could be due to inherent differences in disease features (e.g., parasitemia) or due to true species-specific differences in blood-stage parasite regulation. Several studies have described species-specific gene expression between different species of *Plasmodium* parasites [66,67] but have primarily examined species-specific genes [67] and proteins [66]. While expression of species-specific variant surface antigens [68] and invasion machinery [69] has been documented, particularly for parasites such as *P. falciparum* [70] and *P. vivax* [71], our data suggest that there may also be species-specific expression of genes present in both genomes (i.e., orthologous genes), including genes involved in gametocytogenesis or immune modulation.

We chose here to use CIBERSORTx [30] to estimate the relative proportion of each parasite developmental stage, including sexual stages, present in each infection. In contrast to methods developed to estimate the developmental age of parasites [18,72], which work well on relatively homogeneous parasite populations, this method [45] allows characterization of complex mixtures of stages present in a sample (including the gametocytes), and allows for correction of statistical tests for these proportions. This correction is critical for analyzing parasite RNA-seq data generated directly from blood samples since even rare parasite stages can dramatically impact the overall gene expression profile due to the stage-specific differences in transcriptional activity.

The RNA-seq data generated also enabled a first glance at the genetic diversity of *P. ovale* in Mali using characterization of the DNA polymorphisms present in expressed transcripts. Despite the small sample size (only four *P. ovale* infections analyzed), our study revealed the presence of both sub-species of *P. ovale*. Interestingly, both the parasite and host gene expression profiles of infections caused by these highly divergent parasites were very similar compared to the profiles of *P. falciparum* infections. Indeed, we observed greater variation among the parasite gene expression profiles of *P. falciparum* infections than between those of *P. ovale curtisi* and *P. ovale wallikeri* infections. In addition, we observed comparable levels of genetic

diversity among *P. falciparum* parasites as among *P. ovale curtisi* parasites. This is surprising given the stark difference in prevalence between the two species (and therefore in their population size). While it is difficult to precisely determine the prevalence of *P. ovale*, due to under-detection and species misidentification, *P. ovale* has been reported at about 2% prevalence in Mali compared to ~50% for *P. falciparum* [73]. The observation of similar genetic diversity despite drastic differences in (census) population size is puzzling and suggests that i) the prevalence of *P. ovale* in Mali is widely underestimated, due to misdiagnosis or high proportion of asymptomatic infections, or ii) that there is a large hidden reservoir of *P. ovale* parasites. This observation will require validation using larger cohorts but is important to consider as we move closer towards malaria elimination, as it may indicate that some parasite populations are able to maintain a high level of genetic diversity despite little circulation in the population.

## Conclusions

Here, we described the transcriptional profiles of host and parasites during malaria infections caused by *P. ovale* or *P. falciparum*. We found that host factors contribute more to the human gene expression profiles than the species causing the infection, suggesting i) that age, sex or other individual host characteristics play a key role in determining the regulation of white blood cells during malaria infections, and ii) that the host responses to *P. ovale* and *P. falciparum* infections are not drastically different (for uncomplicated malaria infections). Despite this overall similarity in response, we detected a few human genes differentially regulated in infections with *P. ovale* vs *P. falciparum* suggesting that the host adaptive immune response to these parasites may differ. In addition to insights on the transcriptional regulation of the parasites, this study enabled rigorous characterization of DNA polymorphisms, which revealed the presence of both sub-species of *P. ovale* and a surprisingly high level of genetic diversity in *P. ovale* (comparable to that of *P. falciparum*). Overall, this study provides new insights on the regulation and diversity of *P. ovale* infections that have important implications for the development of pan-malaria vaccines and for developing approaches to eliminate malaria.

## Methods

### Ethics approval and consent

Individual informed consent/assent was collected from all children and their parents. The study protocol and consent/assent process were approved by the institutional review boards of the Faculty of Medicine, Pharmacy and Dentistry of the University of Maryland, Baltimore and of the University of Sciences, Techniques and Technologies of Bamako, Mali (IRB numbers HCR-HP-00041382 and HP-00085882).

### Samples

Samples included in this study were collected from uncomplicated malaria infections from treatment-seeking children from Bandiagara, Mali [28]. Briefly, blood samples were collected from children during unscheduled, patient-initiated visits with i) presentation of symptoms consistent with malaria (fever, headaches, joint pain, vomiting, diarrhea, or abdominal pain) and ii) identification of *Plasmodium* parasites by thick smear. All infections were successfully treated with antimalarial drugs. Whole-blood samples were collected and preserved in PAX-gene blood RNA tubes and stored at -80˚C until extraction [28].

We selected, for these analyses, nine blood samples collected from three children successively infected with *P. falciparum* and *P. ovale* (determined by light microscopy [28]).

### Generation of RNA-seq data

We extracted RNA from whole blood using MagMax blood RNA kits (Themo Fisher) (between 0.24 μg and 3.16 μg total from each sample). Total RNA was subjected to rRNA depletion and polyA selection (NEB) before preparation of stranded libraries using the NEB-Next Ultra II Directional RNA Library Prep Kit (NEB). cDNA libraries were sequenced on an Illumina NovaSeq 6000 to generate ~55–130 million paired-end reads of 75 bp per sample (**S2 Table**). To confirm the *Plasmodium* species responsible for the malaria episode, we first aligned all reads from each sample using hisat2 v2.1.0 [74] to a fasta file containing the genomes of *P. falciparum* 3D7, *P. vivax* PvP01, *P. malariae* UG01, and *P. ovale curtisi* GH01 genomes downloaded from PlasmoDB [75] v55. For the remaining analyses described in this study, we relied on the alignment of all reads using hisat2 to a fasta file containing the *P. falciparum* 3D7, *P. ovale* GH01 and human hg38 genomes i) using default parameters and ii) using (—max-intronlen 5000). Reads mapping uniquely to the hg38 genome were selected from the BAM files generated with the default parameters. Reads mapping uniquely to either *Plasmodium* genome were selected from the BAM files generated with a maximum intron length of 5,000 bp. PCR duplicates were removed from all files using custom scripts. We then calculated read counts per gene using gene annotations downloaded from PlasmoDB (*Plasmodium* genes) and NCBI (human genes) and the subread featureCounts v1.6.4 [76].

### Gene expression analysis

We excluded one sample, B4, from all analyses due to a high percent of duplicated reads (96.1% of human reads, 94.5% of *P. falciparum* reads, **S2 Table**). For all other samples, read counts per gene were normalized into counts per million (CPM), separately for human and *Plasmodium* genes. To filter out lowly expressed genes, only human genes that were expressed at least at 10 CPM in > 50% of the samples were retained for further analyses (9,884 genes). *Plasmodium* genes were filtered using the same criteria, and additionally selected to only include 1:1 orthologs between *P. falciparum* and *P. ovale* (2,631 genes). Read counts were normalized via TMM for differential expression analyses. Statistical assessment of differential expression was conducted, separately for the human and *Plasmodium* genes, in edgeR (v 3.32.1) [77] using a quasi-likelihood negative binomial generalized model i) without covariates for human genes and ii) with and without correcting for proportion of each parasite developmental stage for *Plasmodium* reads. All results were corrected for multiple testing using false discovery rate (FDR) [78].

### Gene expression deconvolution

CIBERSORTx [30] was used to estimate, in each sample, the proportion of i) human immune cell subtypes and ii) *Plasmodium* developmental stages. To deconvolute human reads, we used as a reference LM22 [79], a validated leukocyte gene signature matrix using 547 genes to differentiate 22 immune subtypes (collapsed to eight categories in our analysis). A custom signature matrix derived from *P. berghei* scRNA-seq data was used for *P. falciparum* and *P. ovale* stage deconvolution, using orthologous genes for the appropriate species [45].

### Complexity of infection and genotyping

To assess the complexity of each infection (i.e., monoclonal vs. polyclonal), allele frequency plots [80] were generated for each sample by calculating the proportion of reads with a given reference allele at each nucleotide position covered at > 50X. We also calculated $F_{ws}$ for all *P. falciparum* and *P. ovale* infections using moimix, excluding multi-gene families, according to

the methodology described in Bradwell et al. [23]. Pairwise nucleotide differences were determined using each position covered at > 20X in a given pair of samples, separately for *P. falciparum* and *P. ovale* infections.

## Phylogenetic analysis

We reconstructed the entire *Plasmodium* Cytochrome B sequence from each sample using the mpileup file generated from the RNA-seq data and using, at each nucleotide position covered by at least 20 reads, the allele present in most reads. We then generated a neighbor-joining tree with MEGA 11 [81] using the cytochrome B sequences from all sequenced isolates and publicly available sequences for *P. ovale curtisi*, *P. ovale wallikeri* and *P. vivax* in and 500 bootstraps.

## Supporting information

**S1 Fig. Host genes differentially expressed in *P. ovale* vs. *P. falciparum* infections. (A)** Differences in host gene expression between *P. ovale* and *P. falciparum* infections. Each dot represents a human gene and is displayed according to the log fold-change (x-axis) and -log10 p-value (y-axis) and colored according to the statistical significance (black–non-significant, red–significantly overexpressed in *P. falciparum* infections, blue–significantly overexpressed in *P. ovale* infections, FDR = 0.1). **(B)** Gene expression deconvolution results of infections with *P. falciparum (left)* or *P. ovale (right)*. Chi-square tests were performed for each individual to compare the immune cell composition during *P. falciparum* and *P. ovale* infections. Individual A: $X^2$ = 2.99, p = 0.56, Individual B: $X^2$ = 7.77, p = 0.10, Individual C: $X^2$ = 0.89, p = 0.93
(TIF)

**S2 Fig. Parasite genes differentially expressed in *P. ovale* vs. *P. falciparum* infections. (A, B)** Differences in parasite gene expression between *P. ovale* and. *P. falciparum* infections. Each dot represents a parasite gene and is displayed according to the log fold-change (x-axis) and -log10 p-value and colored according to the statistical significance (black–non-significant, red–significantly overexpressed in *P. falciparum* infections, blue–significantly overexpressed in *P. ovale* infections, FDR = 0.1). The volcano plots show the results without correcting the analyses for stage composition differences **(A)** or after correction **(B)**. **(C)** Gene expression deconvolution results from *Plasmodium* RNA-seq reads during infection with *P. falciparum (left)* or *P. ovale (right)*.
(TIF)

**S3 Fig. Stage composition of *P. falciparum* samples (measured by the proportion of trophozoites) is correlated with overall parasite gene expression profiles estimated by PC2.** The scatterplot shows the estimated proportion of trophozoites present in each *P. falciparum* sample (x-axis) relative to the position of this infection along PC2 of **Fig 2** (y-axis).
(TIF)

**S4 Fig.** Complexity of the *P. falciparum* (A) and *P. ovale* (B) infections. Each plot shows the number of nucleotide positions (y-axis) with a particular reference allele frequency (x-axis, from 0 –all reads supporting an alternative allele, to 100%—all reads supporting the reference sequence allele). Note the U-shape distributions indicating the monoclonality of the infections.
(TIF)

**S5 Fig. Gene expression profiles of *P. ovale* infections. (A)** PCA of human gene expression during infection. **(B)** PCA of parasite gene expression during infection.
(TIF)

**S1 Table. Number of reads mapped to each Plasmodium genome.** Samples with more reads mapping to the P. falciparum genome were assumed to be P. falciparum infections. Samples with more reads mapping to the P. ovale genome were assumed to be P. ovale infections. (Note sample B4 was excluded from analyses because of low quality sequencing data). (XLSX)

**S2 Table. Mapping and quality control data from all samples.**
(XLSX)

**S3 Table. Host gene differential expression.**
(XLSX)

**S4 Table. Parasite gene differential expression.**
(XLSX)

## Acknowledgments

We thank the participants and their families for participating in this study, as well as the community of Bandiagara, Mali.

## Author Contributions

**Conceptualization:** Kieran Tebben, David Serre.

**Data curation:** Kieran Tebben.

**Formal analysis:** Kieran Tebben.

**Funding acquisition:** Kieran Tebben, Mark A. Travassos, David Serre.

**Investigation:** Kieran Tebben, David Serre.

**Methodology:** Kieran Tebben.

**Project administration:** David Serre.

**Resources:** Salif Yirampo, Drissa Coulibaly, Abdoulaye K. Koné, Emily M. Stucke, Ahmadou Dembélé, Youssouf Tolo, Karim Traoré, Amadou Niangaly, Andrea A. Berry, Bourema Kouriba, Christopher V. Plowe, Ogobara K. Doumbo, Kirsten E. Lyke, Mahamadou A. Thera, Mark A. Travassos, David Serre.

**Software:** Kieran Tebben.

**Supervision:** David Serre.

**Visualization:** Kieran Tebben.

**Writing – original draft:** Kieran Tebben.

**Writing – review & editing:** Matthew B. Laurens, Christopher V. Plowe, Kirsten E. Lyke, Shannon Takala-Harrison, Mark A. Travassos, David Serre.

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
