## [Decision Letter · Decision Letter 0]

28 Oct 2022

Dear Miss Tebben,

Thank you very much for submitting your manuscript "Malian children infected with Plasmodium ovale and Plasmodium falciparum display very similar gene expression profiles." for consideration at PLOS Neglected Tropical Diseases. As with all papers reviewed by the journal, your manuscript was reviewed by members of the editorial board and by several independent reviewers. In light of the reviews (below this email), we would like to invite the resubmission of a significantly-revised version that takes into account the reviewers' comments. 

You will noticed that that the reviewers are persuaded of the importance of your study. However, they have raised several points that require consideration and revision. One reviewer has noted that there is no replication of the RNAseq analysis. I view this as a substantive concern. Replication is important as the estimation of P values depends on the between-replicate variance, and replication is good scientific practise in any experiment. As these P values underpin a substantial component of this manuscript, a requirement for resubmission is that the RNAseq analysis is replicated.

We cannot make any decision about publication until we have seen the revised manuscript and your response to the reviewers' comments. Your revised manuscript is also likely to be sent to reviewers for further evaluation.

Sincerely,

Paul O. Mireji, PhD

Academic Editor

Ricardo Fujiwara

Section Editor

You will noticed that that the reviewers are persuaded of the importance of your study. However, they have raised several points that require consideration and revision. One reviewer has noted that there is no replication of the RNAseq analysis. I view this as a substantive concern. Replication is important as the estimation of P values depends on the between-replicate variance, and replication is good scientific practise in any experiment. As these P values underpin a substantial component of this manuscript, a requirement for resubmission is that the RNAseq analysis is replicated.

Reviewer's Responses to Questions

**Key Review Criteria Required for Acceptance?**

**Methods**

-Are the objectives of the study clearly articulated with a clear testable hypothesis stated?

-Is the study design appropriate to address the stated objectives?

-Is the population clearly described and appropriate for the hypothesis being tested?

-Is the sample size sufficient to ensure adequate power to address the hypothesis being tested?

-Were correct statistical analysis used to support conclusions?

-Are there concerns about ethical or regulatory requirements being met?

Reviewer #1: (No Response)

Reviewer #2: The study objectives are clear and conceptually this is a very interesting study. 

The design of the study, in terms of approaches to analysis, is generally appropriate, with the major limitation of sample size - see below

The study population, identification and selection of subjects are not adequately described. There should be an explanation of where and the when the subjects were recruited, whether they were identified through active or passive case detection, and most importantly how the diagnosis of malaria was made (as opposed to other causes of symptoms in children with incidental (asymptomatic) parasitemia). This latter point is particularly important because some of the children were afebrile and had / or had very low parasitemia (Table 1). It is not clear which samples were used for the differential expression analysis of host response

The number of subjects included in the study is very small (n=3) which really makes this a pilot study and I think it is inappropriate to make any generalisations from such a small number of subjects (see further comments below). No power calculations have been presented to indicate what proportion of differentially expressed genes might be expected to be detected with such small numbers of subjects, despite the existence of methods to do this. Generally there is insufficient power to support comments about lack of significant differences in most comparisons

The statistical analyses appear correct, but this cannot compensate for the inadequate sample size

There are no ethical or regulatory concerns

Reviewer #3: Could the authors mention about the time-point of collection?

Were the sequencing depth of approximately 55 - 130 million paired-end reads sufficient to obtain sequence coverage of either transcriptome?

Was it necessary to consider a validation of the transcriptome by RT-qPCR of the phenotype of interest?

What were the statistical considerations when analyzing the uneven replicate data set of your three subjects?

**Results**

-Does the analysis presented match the analysis plan?

-Are the results clearly and completely presented?

-Are the figures (Tables, Images) of sufficient quality for clarity?

Reviewer #1: (No Response)

Reviewer #2: No formal analysis plan is presented, but there is no suggestion that the analysis has been manipulated to alter the interpretation of the results

Results are clearly presented and well-explained. 

Figures are clear and good quality

Reviewer #3: Supplemental Figure 2 panel C, the legend needs editing

**Conclusions**

-Are the conclusions supported by the data presented?

-Are the limitations of analysis clearly described?

-Do the authors discuss how these data can be helpful to advance our understanding of the topic under study?

-Is public health relevance addressed?

Reviewer #1: (No Response)

Reviewer #2: Given the limited sample size, I consider that some of the conclusions are not supported by the data presented. 

There is not a specific paragraph in the discussion addressing limitations of the study, and few limitations have been adequately reported elsewhere

The authors do adequately discuss how the findings can advance understanding and some aspects of public health relevance

Reviewer #3: (No Response)

**Editorial and Data Presentation Modifications?**

Reviewer #1: (No Response)

Reviewer #2: Overall the paper is well written and data well presented

Reviewer #3: (No Response)

**Summary and General Comments**

Reviewer #1: In the manuscript by Tebben et al, the authors characterized host and parasite gene expression profiles during Plasmodium ovale versus Plasmodium falciparum infections to dissect molecular mechanisms at play during malaria infections by different Plasmodium parasites. Accordingly, most studies focus on P. falciparum, other Plasmodium species also cause significant public health issues and there is a lack of specific knowledge about these parasites. Additionally, other Plasmodium species, such as P. ovale, typically lead to less severe forms of the disease but little is known about the molecular mechanisms of those non-falciparum infections. In this manuscript, the authors attempted a rare approach of sequencing both host and parasite RNA from nine children infected with either P. falciparum or P. ovale parasites. 

They found that in uncomplicated symptomatic infections, host gene expression profiles differ dramatically according to the participant age rather than by the infecting parasite species. However, parasite gene expression profiles differ between Plasmodium species. They also show that the levels of parasite genetic diversity analyzed using DNA polymorphisms from RNA-seq reads were comparable between Plasmodium falciparum and ovale, despite a dramatic difference in prevalence.

 Although the number of analysed samples is modest, the manuscript is informative and well presented, and particularly interesting for the understudied P. ovale species. Nonetheless, some precisions and analyses are required before publication. 

Comments:

- More epidemiological data is needed on the collected blood samples. Including: symptoms in each patient, date of sample collection (to infer time in between subsequent infections) and season (wet/dry) of sample collection. In Table 1, some individuals have a body temperature < 37.5°C. What was the definition of ‘uncomplicated malaria’ ? Were patients tested for other co-infections (such as helminth)? Sample B3 is not feverish and parasitaemia is more typical of an asymptomatic infection. Why is it classified as uncomplicated malaria?

- Line 410-413: Only reads mapping to the corresponding target genome (hg38, P. falciparum P. ovale) were selected. What is the proportion of reads mapping to both falciparum and ovale why do you exclude them? Unless I misunderstand the methods, it sounds like reads that map to both falciparum and ovale are discarded. 

- In general, Methods should be more detailed. Were samples frozen before RNA extraction? What RNA quantity was recovered in each sample?

- The lack of replicates is a limitation of the study, this should be clearly stated. 

- Line 431: Gene expression deconvolution. The rodent P. berghei scRNAseq data was used to determine the developmental age of each sample. That species does not sequester and, unlike falciparum, all stages are circulating in the blood. As mentioned on line 232, there is a risk that the stage composition is biased. In my experience, the method developed by Lemieux et al (2009 PNAS, Newbold lab) is more accurate, I’m happy to share a R script to determine the parasite developmental age based on the microarray timecourse (data Bozdech 2003). Also, are blood smears available? What was the proportion of ring/troph/gametocytes on these blood smears? 

The comparison of falciparum versus ovale transcriptomes (Fig 2) only makes sense if all samples are at the exact same developmental stage (if not, a normalizing step should be added , similar to Thomson-Luque2021 Nature Communications, Portugal lab) 

- Line 439: Complexity of infection. Supp Fig 4 is not very convincing that all samples are monoclonal. I recommend calculating the Fws metric on all samples (Auburn 2012 Plos One)

- There is no coexpression analysis combining both datasets (host & parasite) together, as in Lee et al 2018 Science Trans Med. The findings from that paper (human differential gene expression driven by parasite load) should be discussed with the results presented here. 

Side note: in the References section, Lee 2018 is cited on BioRxiv rather than the final publication.

- In most figures, only 8 samples are presented instead of the 9 listed in Table 1. 

- Line 175-182. Among the 127 host DE genes, are genes associated with inflammation and what is the direction of regulation (up or down-regulated) according to the two plasmodial species? 

- Line 273: delete "(" at the beginning of the sentence

- Supplementary Figure 3 is unclear and is not mentioned in the main text. 

- I probably did something wrong, but I couldn’t retrieve the Bioproject ‘PRJNA878485’ ?

Reviewer #2: This study is quite innovative, trying to compare the blood transcriptome (of host and parasite) between P falciparum and P ovale infections. P ovale is relatively neglected in terms of research, and difficult to study because it is much less common than P falciparum and often exists as co-infections. Therefore the comparison of P falciparum and P ovale monoinfections in this way is novel and the authors have undertaken quite extensive analysis to investigate both host and parasite biology. Studying these during sequential mono-infections in the same subjects is a novel approach but unfortunately it is very hard to collect such samples and so only 3 subjects are included in the study. 

My biggest concern is that the subjects are described as having "symptomatic malaria" but the evidence presented in Table 1 does not strongly support this. Several of the infection episodes are associated with lack of fever, and in some cases the parasitemias are very low. No mention is made of the case definition of "symptomatic malaria" and there is no report of what other microbiological, virological, and parasitological investigations were done to rule out alternative sources of fever, which will be common in children of this age in Mali. The onus is on the authors to convince us that these children really did all have symptomatic malaria, without co-infections, and that parasitemia was not simply an incidental finding during some of these episodes. This will potentially have a major impact on the interpretation of all subsequent analysis of the host response and possibly the parasite transcriptome

My next major concern is that the sample size is extremely small. Presumably this small sample size precludes adjustment for leukocyte mixture in the analysis of host gene expression? Nevertheless 127 differentially expressed host genes were found between P ovale and P falciparum infections. The authors state that this confirms the overall similarity of the host response to these infections. However I consider it quite remarkable that with this small number of subjects it was possible to detect so many differentially expressed genes, and that with larger sample sizes there could easily be a much larger number of differentially expressed genes. This can potentially be resolved by performing a formal power calculation to understand the proportion of differentially expressed genes which might be detected with a sample size of just 3 subjects. Similarly the authors suggest that the differential expression is not due to differences in leukocyte proportions in blood, based on Chi-squared test P>0.1. Again they should perform a power calculation to determine what magnitude of difference in proportions of cell types they would be able to detect at a significance threshold of P<0.1 (or P<0.05). I suspect they would only be able to detect extremely large differences with confidence. 

The approach of showing the variance in gene expression explained by species, age, parasitemia and individual is quite nice, but as the authors acknowledge out, age and individual are are not independent, and no account has been taken of the fact that parasitemia and parasite species are also not independent of one another - typically P vivax infections have lower parasitemia than P falciparum. Therefore this analysis is unfortunately flawed, and would need to be modified to account for this. 

The use of orthologous genes to estimate parasite developmental stage in analysis of parasite gene expression is good, but the authors should mention that the stage specific gene expression of P ovale is not as well known as P falciparum, so it may not be as accurate. Similar to my comment above about the human gene expression, the interpretation of parasite gene expression also needs to be tempered by the statistical power of the analysis: "Accounting for stage composition differences, only 118 orthologous genes remained differentially expressed between parasite species". 118 differentially expressed genes actually sounds like a lot for such a small sample size.

In light of these comments there is a need to undertake some substantial reanalysis and revision of the title, results and discussion text accordingly. 

Minor comment:

Gene symbols need to be corrected to standard nomenclature for human genes ie all Upper Case.

Reviewer #3: The authors would need to validate their transcriptome via RT-qPCR.

PLOS authors have the option to publish the peer review history of their article (what does this mean?). If published, this will include your full peer review and any attached files.

Reviewer #1: Yes: antoine claessens

Reviewer #2: No

Reviewer #3: No
---

## [Decision Letter · Decision Letter 1]

16 Jan 2023

Dear Miss Tebben,

We are pleased to inform you that your manuscript 'Malian children infected with Plasmodium ovale and Plasmodium falciparum display very similar gene expression profiles.' has been provisionally accepted for publication in PLOS Neglected Tropical Diseases.

Best regards,

Paul O. Mireji, PhD

Academic Editor

Ricardo Fujiwara

Section Editor

Please address the pending concerns by the reviewers in your submission.

Reviewer's Responses to Questions

**Key Review Criteria Required for Acceptance?**

**Methods**

-Are the objectives of the study clearly articulated with a clear testable hypothesis stated?

-Is the study design appropriate to address the stated objectives?

-Is the population clearly described and appropriate for the hypothesis being tested?

-Is the sample size sufficient to ensure adequate power to address the hypothesis being tested?

-Were correct statistical analysis used to support conclusions?

-Are there concerns about ethical or regulatory requirements being met?

Reviewer #1: (No Response)

Reviewer #2: This is a review of revised manuscript

The authors have updated all of the methods appropriately

Reviewer #3: The absence of a robust sample size is a fatal flaw that compromises the statistical power to test the hypothesis.

**Results**

-Does the analysis presented match the analysis plan?

-Are the results clearly and completely presented?

-Are the figures (Tables, Images) of sufficient quality for clarity?

Reviewer #1: (No Response)

Reviewer #2: This is a review of revised manuscript

The authors have modified the results to address the reviewer comments and all changes are appropriate. My comments have been adequately addressed

Reviewer #3: (No Response)

**Conclusions**

-Are the conclusions supported by the data presented?

-Are the limitations of analysis clearly described?

-Do the authors discuss how these data can be helpful to advance our understanding of the topic under study?

-Is public health relevance addressed?

Reviewer #1: (No Response)

Reviewer #2: The conclusions have been revised with appropriate caveats. All of my previous concerns have been addressed

Reviewer #3: (No Response)

**Editorial and Data Presentation Modifications?**

Reviewer #1: (No Response)

Reviewer #2: None

Reviewer #3: (No Response)

**Summary and General Comments**

Reviewer #1: Overall, the authors have correctly addressed my comments and have improved the manuscript. I recommend its publication.

There was a slight misunderstanding when I mentioned the lack of replicates, I didn’t mean that qRT-PCR was needed (I completely agree with the authors on that point), I meant that biological/technical replicates of the RNAseq had not been performed (See Tarr2018 (https://doi.org/10.1186/s12864-018-5257-x ) for an in-depth analysis of replicates of P.f. RNAseq data). My apologies for the imprecise wording. I think the paper is now clear on that point.

The estimation of the developmental age is now well explained. It relies on the published CIBERSORT software, even though the details are currently unpublished (Ref 45 is not on BioRxiv).

This is outside the scope of this paper, but I would be curious to see the same analysis (performed for Fig Supp 2C) on the Tonkin-Hill2018 and Andrade2020 datasets…

Another explanation as to why so many P.f. troph/schizont sequencing reads are present in these samples may be because whole blood samples were used, as opposed to purified red blood cells. It has been shown repeatedly that plasma contains a substantial amount of Plasmodium DNA and protein, and that this DNA is more representative of total parasitaemia (as opposed to circulating rings and sequestered troph/schizonts). See for example Imwong 2014 JID (https://doi.org/10.1093/infdis/jiu590). I do not know of a study measuring the amount of Plasmodium RNA in human plasma, but it seems reasonable to assume that there is some.

Line 94: indicate that the study was done on P. ovale curtisi (ref 27)

Reviewer #2: The authors have addressed all of my concerns about the initial manuscript to my satisfaction

Reviewer #3: (No Response)

PLOS authors have the option to publish the peer review history of their article (what does this mean?). If published, this will include your full peer review and any attached files.

Reviewer #1: **Yes: **Antoine Claessens

Reviewer #2: No

Reviewer #3: No

---

## [Editor Report · Acceptance letter]

20 Jan 2023

Dear Dr. Serre,

We are delighted to inform you that your manuscript, "Malian children infected with Plasmodium ovale and Plasmodium falciparum display very similar gene expression profiles.," has been formally accepted for publication in PLOS Neglected Tropical Diseases.

Best regards,

Shaden Kamhawi

co-Editor-in-Chief

Paul Brindley

co-Editor-in-Chief
